# Age-Associated Salivary MicroRNA Biomarkers for Oculopharyngeal Muscular Dystrophy

**DOI:** 10.3390/ijms21176059

**Published:** 2020-08-22

**Authors:** Vered Raz, Rosemarie H. M. J. M. Kroon, Hailiang Mei, Muhammad Riaz, Henk Buermans, Saskia Lassche, Corinne Horlings, Bert De Swart, Johanna Kalf, Pradeep Harish, John Vissing, Szymon Kielbasa, Baziel G. M. van Engelen

**Affiliations:** 1Department of Human Genetics, Leiden University Medical Centre, 2333ZC Leiden, The Netherlands; muhammad.riaz@yale.edu (M.R.); henk.buermans@10xgenomics.com (H.B.); 2Radboud University Medical Center, Department of Rehabilitation, Donders Institute for Brain, Cognition and Behaviour, 6525AJ Nijmegen, The Netherlands; Rosemarie.Kroon@radboudumc.nl (R.H.M.J.M.K.); Bert.deSwart@Radboudumc.nl (B.D.S.); Hanneke.Kalf@radboudumc.nl (J.K.); 3Sequence Analysis Support Core, Leiden University Medical Centre, 2333ZC Leiden, The Netherlands; H.Mei@lumc.nl (H.M.); S.M.Kielbasa@lumc.nl (S.K.); 4Department of Neurology, Donders Institute for Brain, Cognition and Behaviour, Radboud University Medical Centre, 6525AJ Nijmegen, The Netherlands; Saskia.Lassche@radboudumc.nl (S.L.); Corinne.Horlings@radboudumc.nl (C.H.); Baziel.vanEngelen@radboudumc.nl (B.G.M.v.E.); 5Centre of Gene and Cell Therapy, Royal Holloway, University of London, Egham TW2 0EX, UK; Pradeep.Harish@rhul.ac.uk; 6The Copenhagen Neuromuscular Center, Righospitalet, University of Copenhagen, DK-2100 Copenhagen, Denmark; john.vissing@regionh.dk

**Keywords:** OPMD, miRNA, muscle aging, muscle atrophy, circulating miRNA

## Abstract

Small non-coding microRNAs (miRNAs) are involved in the regulation of mRNA stability. Their features, including high stability and secretion to biofluids, make them attractive as potential biomarkers for diverse pathologies. This is the first study reporting miRNA as potential biomarkers for oculopharyngeal muscular dystrophy (OPMD), an adult-onset myopathy. We hypothesized that miRNA that is differentially expressed in affected muscles from OPMD patients is secreted to biofluids and those miRNAs could be used as biomarkers for OPMD. We first identified candidate miRNAs from OPMD-affected muscles and from muscles from an OPMD mouse model using RNA sequencing. We then compared the OPMD-deregulated miRNAs to the literature and, subsequently, we selected a few candidates for expression studies in serum and saliva biofluids using qRT-PCR. We identified 126 miRNAs OPMD-deregulated in human muscles, but 36 deregulated miRNAs in mice only (pFDR < 0.05). Only 15 OPMD-deregulated miRNAs overlapped between the in humans and mouse studies. The majority of the OPMD-deregulated miRNAs showed opposite deregulation direction compared with known muscular dystrophies miRNAs (myoMirs), which are associated. In contrast, similar dysregulation direction was found for 13 miRNAs that are common between OPMD and aging muscles. A significant age-association (*p* < 0.05) was found for 17 OPMD-deregulated miRNAs (13.4%), whereas in controls, only six miRNAs (1.4%) showed a significant age-association, suggesting that miRNA expression in OPMD is highly age-associated. miRNA expression in biofluids revealed that OPMD-associated deregulation in saliva was similar to that in muscles, but not in serum. The same as in muscle, miRNA expression levels in saliva were also found to be associated with age (*p* < 0.05). Moreover, the majority of OPMD-miRNAs were found to be associated with dysphagia as an initial symptom. We suggest that levels of specific miRNAs in saliva can mark muscle degeneration in general and dysphagia in OPMD.

## 1. Introduction

Following improvements in RNA technologies, the subclass of small RNAs, microRNAs (miRNA), have been proposed as a diagnostic and therapeutic tool. miRNAs are small, 18–25 nucleotides, that play a regulatory role in mRNA degradation, and hence affect mRNA expression profiles [1]. Therefore, deregulated miRNAs have been targeted for clinical applications [2,3]. miRNAs are highly stable, and as such, they are suitable as biomarkers and diagnosis [2,3,4]. Since miRNA is extracellularly secreted [5], it can be detected in circulating biofluids using quantitative procedures. The combination of those features can potentially make miRNAs excellent biomarkers, and large-scale studies were performed to identify miRNA candidates in various pathological conditions and to implement them for clinical use [2,3,4]. miRNA can be quantitatively measured, and therefore can accurately and objectively report changes in conditions. miRNAs often have high sensitivity, and therefore can indicate early stages of a disease. miRNAs often have high sensitivity, and therefore can indicate early stages of a disease [4]. It has been suggested that miRNA biomarkers can be useful for clinical decision-making, tailoring of therapy, and predicting clinical outcomes [6]. Molecular biomarkers have been suggested as beneficial to monitor disease progression, especially in chronic conditions and/or in age-associated disorders [7,8]. Measuring miRNA levels during disease progression could be crucial for clinical outcomes, especially in conditions where symptoms are monitored by subjective clinical measurements [6]. Therefore, predictive and diagnostic molecular biomarkers are already used in treatment decisions for various cancers [4]. The identification of reliable biomarkers and, consequently, their implementation in the clinic are lacking for the rare disorders, which includes the majority of neuromuscular disorders (NMDs).

Muscle weakness in many adult NMDs slowly progresses with age. Clinical outcomes are made by laborious and subjective neurological tests. As the first step to improve predictions of clinical outcome research, miRNAs as potential biomarkers have been investigated for several NMDs [9,10,11,12,13,14]. Identifying miRNAs as biomarkers for NMDs is especially challenging because many of the adult NMDs are rare. miRNAs for oculopharyngeal muscular dystrophy (OPMD) have not been reported.

OPMD is a rare adult-onset myopathy (estimated prevalence worldwide 1:100,000; onset age from 45 years onwards), with a slow age-associated progression [15]. The initially affected muscles in OPMD include eyelid muscle weakness leading to ptosis, and pharyngeal muscle weakness leading to dysphagia [15]. Despite these clear clinical characteristics, OPMD is considered as under-diagnosed [16,17]. OPMD is a monogenic, autosomal, mostly dominantly inherited disease, caused by an expansion of mutation in the gene encoding for PABPN1 [15]. Lacking reliable and significant biomarkers for OPMD could hamper the development of therapies for this condition [18,19].

Here we report the identification of miRNA biomarkers for OPMD, under the hypothesis that miRNAs whose expression levels are specifically altered in muscles from OPMD patients would also changed in biofluids of OPMD patients. Our study design was divided into two parts (Figure 1): first identifying miRNAs that are differentially expressed in OPMD muscles using deep RNA sequencing. For this part, we compared the OPMD-miRNAs between human OPMD patients and the mouse model for OPMD. Subsequently, we selected miRNAs candidates using comparative studies, and we determined expression levels in biofluids from OPMD patients using RT-qPCR. We show that the expression of miRNAs biomarkers for OPMD is correlated with age, and with OPMD initial symptoms.

## 2. Results

### 2.1. Small RNAseq in OPMD Patients and the OPMD Mouse Model Revealed Limited Similarities between Differentially Expressed miRNAs

To identify miRNAs that are specifically differentially expressed in OPMD, RNAseqs of small RNA species were carried out in limb muscles from both human vastus lateralis (VL) (Figure 1). In previous studies, we reported that the VL in OPMD patients is both histologically and molecularly affected [20,21]. Since the mouse model for OPMD, A17.1, showed muscle pathology that is causatively associated with the expression of expanded PABPN1 (17 alanine stretch) [22], we also studied the deregulation of miRNAs in this mouse model. In this mouse model, muscle pathology and molecular progression were reported between 6 and 18 weeks in the tibialis anterior (TA) of the A17.1 OPMD mouse model. Therefore, the study here was carried out on mice that were 12 weeks old. Differential expression in humans was carried out between OPMD and age-matching control groups and in mouse, between FVB (Friend Virus B laboratory mouse) control and A17.1 mouse model (Figure 1). As we aimed to compare between the two studies, all wet lab and sequencing procedures were carried out in a single experiment (library prep and sequencing) for both human and mouse samples. RNAseq analysis was carried out using the same pipeline. Quality control features showed that RNAseq in humans had a higher number of reads and a higher percentage of aligned reads compared with the mice samples (Appendix A). The minimum number of aligned reads was above 3 million across all samples (Appendix A). After removal of miRNAs with an average lower than 0.4 cpm (counts per million), 458 and 453 miRNAs remained for differential expression analysis in the humans or mice study, respectively. A principal component analysis (PCA) indicated robust differences between OPMD or the OPMD mouse model and controls in both humans and mice (Appendix A). In humans, 125 miRNAs were identified as differentially expressed (pFDR < 0.05 correction), but in mice, only 36 miRNAs were differentially expressed (Figure 2A,B). From those, the majority (59.5%, 56.4%) were upregulated in OPMD or the mouse model, respectively (Figure 2C).

To compare the two studies, we first determined miRNA overlap using miRNA sequence in humans and in mice. Only 55.9% of the miRNAs in humans overlapped with the mice miRNAs, from which close to 30% were found as differentially expressed (Figure 2D). In the mouse model, however, only 10% of the overlapping miRNAs were found as differentially expressed (Figure 2D). Considering all the overlapping miRNAs, the fold-change direction between humans and mice showed high correlation (*r* = 0.45), but only 15 miRNAs showed significant deregulation (pFDR < 0.05) in both studies (Figure 2E). Only four miRNAs showed opposite dysregulation direction, and eleven miRNAs showed similar dysregulation direction between humans and mice (Figure 2E). Among those, we recognized miR-133, which is often affected in different muscular dystrophies [(11)]. The majority of the significantly deregulated miRNAs had an FC > 2 (Figure 2F). Together, this comparative study reveals that the OPMD-deregulated miRNAs predominantly differ between humans and the mouse model. Therefore, all subsequent analyses were carried out on humans only.

### 2.2. Characteristics of the OPMD-Deregulated miRNAs in Humans

To further characterize features of the OPMD-deregulated miRNA in humans, we compared fold-change to expression levels with the goal to select candidates for biomarkers in OPMD. The majority of the significantly differentially expressed miRNAs showed only mild or low expression levels (Figure 3A). The most highly expressed miRNAs in our study (Figure 3A) were previously reported as myogenic miRNAs (MyomiRs), describing a group of miRNAs that are enriched or specifically expressed in muscles and often are involved in myogenesis (strained or skeletal) [10,11,23].

Since many studies have been reported on miRNAs in diverse muscle pathology conditions, we compared the deregulation direction between OPMD and reported miRNA, focusing on miRNAs that were found in our RNAseq. Only the dysregulation direction was compared as fold changes are affected by normalization and detection procedures. Eight MyomiRs have been reported [24], but only the miR-133 family (mir-133a and miR-133b) were found to be deregulated in our study. In OPMD, the miR-133 family was downregulated (Figure 3B; Appendix A), whereas in other muscular dystrophies they were found to be upregulated [9]. miR-206 showed a trend with higher levels in OPMD (pFDR 0.06) (Figure 3B). miR-206 is also upregulated in other muscular dystrophies [9]. miR-1, a common muscular dystrophy biomarker [9], was not significantly deregulated in OPMD (Figure 3B). In contrast, the mir-29 family is down-regulated in muscular dystrophies [25] and was also down-regulated in OPMD (Appendix A.

We then compared the dysregulation direction of additional miRNA that has been reported to be associated with muscle pathologies. Muscle atrophy and muscle wasting have been reported in OPMD mouse models [22,26,27]. Nine of the reported muscle atrophy-associated miRNAs [28,29,30,31] were found to be significantly deregulated in OPMD, but only five showed a similar dysregulation direction (Figure 3B, Appendix A). OPMD is also aging-associated, and 13 reported muscle aging-associated miRNAs [32,33] were found to be deregulated in OPMD, from which 12 showed a similar dysregulation direction (Figure 3B; Appendix A). We then compared the OPMD-miRNAs to miRNAs of inclusion body myositis (IBM), as IBM is also an aging-associated myopathy, sharing, in part, a similar muscle involvement pattern with OPMD [11,12,34]. All five IMB-miRNAs were found to have a similar dysregulation direction in OPMD (Figure 3B and Appendix A). Together, this suggests that the OPMD-miRNAs share more similarity with muscle aging and aging-associated muscular disorders as compared with other muscular dystrophies.

As OPMD is age-associated, we investigated an age-association to the OPMD-deregulated miRNAs. A total of 17 out of the 125 OPMD-deregulated miRNAs (13.4%) had a significant correlation with age (Appendix A). In contrast, among the non-OPMD-deregulated miRNAs, only 1.4% showed an age-association (Appendix A, 6 out of 419 miRNAs). Not a single miRNA showed an association with sex. This suggests that those miRNAs could mark age-association in OPMD. 

Among the OPMD-deregulated miRNAs, miR-146b, miR-26b and let7a were reported as upregulated in aging [33,35], and miR-133a was reported as downregulated in aging (Appendix A). Although the direction of correlation was similar between aging and OPMD, a significant correlation was found only for miR-146b (Figure 3C). From these OPMD-deregulated and age-associated miRNAs, 71% (12 out of 17) showed a positive correlation between age and expression levels (Figure 3C). Moreover, for 11 miRNAs (65%) the age-association was stronger in OPMD compared with control samples (Figure 3C; Appendix A). Age-associated expression levels in OPMD and control samples are shown for three miRNAs examples (Figure 3D–F). For all three examples, the slope of the linear regression model for age association was higher in OPMD compared with control (Figure 3D–F). This indicates that age-associated changes are larger in OPMD compared with control. miR-459 showed that age-association exhibits a positive trend that was not significant in controls (Figure 3D). In contrast, an age-association was not found in miR-146b or miR-200c, control group (Figure 3E,F, respectively). The miR-146b has been reported as age-associated in a muscle aging study [33]. However, in that study, aging was investigated between two age groups, young (mean 31 years old) and old (mean 73 years old). In our study, the youngest age was 43, and in our regression model, age was a continuous range, which could explain the differences in results between the two studies. 

miRNA binding to a specific sequence in mRNA targets this mRNA to the mRNA degradation machinery, resulting in reduced expression levels [36]. Since a single miRNA has multiple mRNA targets, bioinformatic tools have been developed predicting target genes [37]. We previously published the OPMD-transcriptome in VL muscles [20]. Here, we overlapped target genes to the OPMD-miRNA with the OPMD-deregulated mRNAs. Since miRNA upregulation leads to mRNA downregulation, we focused on the OPMD-upregulated miRNAs with an average CPM higher than 2.5, and a fold-change higher or equal to 2. In total, 11 miRNAs met these criteria (Figure 4). For only seven miRNAs (Figure 4), target genes are reported in the miR database (mirdb.org). After overlapping the predicted target genes with the OPMD-downregulated genes, 70 were identified (Figure 4; Appendix A). We then repeated the identification OPMD-target genes procedure for 11 OPMD miRNAs whose expression levels were positively age-associated (pFDR < 0.05) (Figure 4). In total, 85 OPMD-downregulated predicted targets were found for the OPMD-upregulated and age-associated miRNAs (Figure 4; Appendix A). In a pathway enrichment analysis for each gene group, only the ubiquitin conjugation pathway was found as significant, (pFDR < 0.05) (Figure 4). Interestingly, only three miRNAs overlapped between the two lists, but 43 (53%) OPMD gene targets overlapped between the two lists, and those genes were also enriched in the ubiquitin conjugation pathway (Figure 4). The ubiquitin proteasome system (UPS) was found to be the most significantly deregulated in OPMD, and prediction models suggested a high correlation between expression levels of deregulated UPS genes in OPMD [22,26,38,39]. Among the target genes, we recognized *ARIH2* (Appendix A), an E3 ligase whose expression levels are regulated by PABPN1 [40]. This suggests that the genes of the ubiquitin proteasome system (UPS) are significantly affected by OPMD-miRNAs and specifically those that are age-associated.

### 2.3. Assessment and Verification of miRNA Expression in Biofluids

Aiming to identify biomarkers in biofluids that could indicate muscle weakness in OPMD, we investigated expression levels of the OPMD-deregulated miRNAs in serum and saliva. We first conducted RNAseq in VL muscles and saliva from OPMD patients, and correlated expression levels in both tissues (Figure 1). We chose to focus on miRNAs that highly correlated between the two tissues for further studies. Muscle and saliva were collected from the same subjects allowing paired analysis. RNA isolation and library prep were conducted in a single experiment in order to reduce technical variations. Overall, miRNA expression levels were much lower in saliva than in muscles (Appendix A). Expression levels did not correlate between the two tissues for the majority of miRNAs, but for 6% of the detected miRNAs (67 miRNAs) the correlation was significant and positive (Appendix A). 

We then selected candidates for verification and expression studies using the following criteria: first, OPMD-deregulated in vastus lateralis (pFDR < 0.05) and absolute fold-change above; second, miRNAs with expression levels that highly correlate between muscle and saliva. In total, nine OPMD-deregulated miRNAs were selected (Figure 5). Five miRNAs with high expression levels in saliva but unchanged between OPMD and control VL muscle were included as control. Expression levels were determined using qRT-PCR. Serum and saliva were collected from 33 OPMD patients, and 18 controls (Figure 1). The control group for this study consisted of some of the OPMD spouses. Biofluid collection was carried out with the same protocol for all subjects. Two spike miRNAs were used to monitor technical variations during the procedure, and to compare between serum and saliva. CT values for both spikes were consistent across all samples and did not differ between serum and saliva (Appendix A). After normalization to the average of two spike miRNAs, the average expression levels were found to be consistently lower in serum compared with saliva for 14 miRNAs, except for miR-451a (Figure 5A and Appendix A). Five miRNAs in serum and four miRNAs were excluded because their expression (CT values) were found in only half or less of the subjects (Appendix A). In addition, miR-133b expression levels in serum were under the detection limit mainly in the control group and therefore it was also excluded (Figure 5B). Together, the confidence in results obtained from saliva would be higher compared with serum.

Differences in expression levels between OPMD and control groups were found significant in saliva for five miRNAs and in serum for seven miRNAs (Figure 5B). In general, the same fold-change direction between muscle and saliva was found for four miRNAs (Figure 5C), whereas in serum, the fold-change direction was opposite to that in muscles (Figure 5D). None of the OPMD-upregulated miRNAs were detected in serum, and higher expression levels were found for both the OPMD-downregulated miRNAs and the unchanged miRNAs in muscles (Figure 5D). Levels of miR-200c were higher in OPMD in both muscles and saliva, and lower levels of mir-15a-5p, miR-29c-3p and let7i were measured in both OPMD saliva and OPMD muscles (Figure 5B,C). miR-451a, however, had unchanged levels in muscles and lower levels in OPMD saliva compared with the control group (Figure 5C).

We then continued the selection for the most potent miRNA biomarker(s) for OPMD and we investigated an age-association for all differentially expressed miRNAs in both serum and saliva. None of the differentially expressed miRNA in serum showed an age-association (Appendix A). In contrast, the expression of miR-200c and miR-451 in saliva was found as age-associated in OPMD (Figure 5E,F). The slope of the regression line indicated higher expression levels with age (Figure 5E,F). Last, we assessed if miRNAs levels are significantly changed with the initial symptoms, ptosis or dysphagia. In general, around half of the patients had either ptosis or dysphagia as initial symptoms, and in only two patients, leg weakness was noted as initial symptoms (Appendix A). Levels of most OPMD-miRNAs were significantly changed in patients with dysphagia (Figure 5G,H). Only levels of miR-451a in saliva were significantly changed in patients with ptosis (Figure 5G). Changes in let-7i levels in saliva were not specific to the initial diagnosis (Figure 5G). In sum, four miRNAs in either saliva or serum showed a correlation with dysphagia, but only two miRNAs in salvia and none in serum were associated with ptosis, as an initial symptom (Figure 5G).

## 3. Discussion

To identify miRNAs biomarkers for OPMD, an age-associated adult myopathy, we conducted a comparative RNAseq study in human- and mouse-affected muscles. We report only small similarities between human and mouse deregulated miRNAs, which agrees with our previous study showing limited similarities between deregulated cytokines in OPMD and A17.1 mouse model [41]. Under the hypothesis that miRNAs in affected OPMD muscles could be found in circulating biofluids, and therefore would be OPMD-specific, we studied the expression of selected candidates in both saliva and serum from OPMD patients. Our results show that, in serum, the OPMD-deregulated miRNAs tend to have a higher fold-change (OPMD vs. control) compared with saliva. However, we show here that miRNA expression levels in serum are generally lower than in saliva. We, therefore, suggest that expression values in saliva might be more reliable compared with serum. Moreover, as fold-change direction in saliva, but not in serum, could be the same as in muscle, we suggest that saliva is a more suitable biofluid to measure levels of miRNAs that indicate muscle weakness in OPMD. Future studies should also investigate miRNAs as biomarkers in saliva for other myopathies.

Consistent with an age(ing)-association of OPMD symptoms, we found that a larger number of the OPMD-miRNA were age-associated as compared with control. Moreover, the changes in expression levels with age were greater in OPMD compared with control. The OPMD-miRNAs that were found to be age-associated could potentially report disease progression; however, this must be investigated in a longitudinal study. Our analysis further showed that more OPMD-deregulated miRNAs are associated with dysphagia rather than ptosis initial symptoms. This observation could be expected for miRNA measurements in saliva, but not necessarily in serum. For the significantly deregulated OPMD-miRNAs, the fold-change direction in saliva is similar to that in affected muscles, but in serum, it is the opposite. Last, expression levels for two of the OPMD-deregulated miRNAs in saliva, miR-200c and miR-451, correlated with age. None of the OPMD-deregulated miRNAs were found as age-associated. Only miR-200c was found to be upregulated in both OPMD-affected muscles and in saliva from OPMD patients. miR-200c expression levels in saliva were found to be higher than miR-451, and its expression was specifically high in patients with dysphagia as an initial symptom. Together, miR-200c could be a relevant biomarker for OPMD. The prognostic values of these miRNAs as biomarkers for disease progression should be verified in follow-up longitudinal studies where samples and clinical development are assessed over time. 

Several studies demonstrated therapeutic potential for deregulated miRNAs in NMD animal models [42,43,44]. A recent study suggested a role for miR-200c in muscle wasting and myogenesis [45]. Moreover, miR-200c was suggested to protect neurons from damage induced by protein aggregation [46]. Protein aggregation, muscle wasting and reduced myogenesis have been reported in OPMD models [20,22], and future studies should investigate if miR-200c plays a causative role in OPMD. Our study suggests that miR-200c can be a predictive biomarker for OPMD.

Circulating miRNAs have been reported to indicate disease progression for several neuromuscular disorders [13]. Amongst these, the myomiRs have been reported to be upregulated in several muscle dystrophies [47]. Our comparative analysis here showed that only few of the myomiRs are deregulated in OPMD, and that there is a downregulation of the deregulated. In contrast, the OPMD-miRNAs were found to be most consistent with aging miRNAs. This observation is consistent with our previous study showing that the OPMD transcriptome was significantly associated with muscle aging transcriptomes [20]. In addition, a similar dysregulation direction was found in patients with IBM. Both myopathies share pathological hallmarks including vacuoles in muscle histology and dysphagia as clinical phenotype. The similarity in the deregulated miRNAs could indicate similar underlying molecular and pathophysiological mechanisms. Moreover, *PAX7* upregulation and an increase in satellite cells was reported in both OPMD and IBM muscles [48,49]. 

Our assays showed high interindividual variations in miRNA expression levels between OPMD patients, and in a few patients, the expression level of specific miRNAs was not different from that found in the control. Among the Dutch OPMD patients symptom severity highly varied and the initial symptom as well [50,51]. It could be expected that few miRNAs would be required as biomarkers for OPMD. This is the first miRNA study for OPMD, and additional studies are required in order to verify and sustain the results found in this study.

## 4. Material and Methods

### 4.1. Muscle Tissue and Biofluid Collection

The muscles from OPMD patients and control subjects were previously reported in [20,43,44]. The vastus lateralis muscle biopsy was collected with a Bergstrom needle biopsy and snapped freeze in liquid nitrogen. Collection of saliva and blood from OPMD patients and control subjects was reported in [51]. Saliva was collected in RNeasy Protect Saliva Mini Kit (Qiagen). Blood was collected in EDTA collection tubes, and after centrifugation, the serum was collected and stored. Inclusion criteria:Only subjects older than 18 years were included.OPMD patients were clinically diagnosed by the neurologist based on the common clinical OPMD symptoms. OPMD was genetically confirmed using a genetic test as described in [51]. Patients with OPMD-like symptoms without a genetic confirmation were excluded.Healthy controls were the partners of the OPMD patients.

Both studies were approved by the Regional medical ethics committee (CMO nr. 2005/189 for the study in muscles; and CMO nr. NL54606.091.15 for the study in serum and saliva). A written informed consent was obtained from all participants. The sample size for the muscle study here is based on the OPMD transcriptome study [20]. The sample size for the saliva and serum studies were calculated as 2.5 fold of the size of the muscle study.

The tibialis anterior muscle was collected from A17.1 and FVB 12 weeks old male mice, as described in [41]. The sample size for the mouse muscle study is based on the transcriptome study in this mouse model 

All muscles were stored at −80 °C for ex-vivo analyses. A summary of controls, patients and mouse samples are found in Appendix A.

### 4.2. RNAseq Analysis

The paired-end raw reads were aligned to the GRCm38/mm10 mouse reference genome using the BIOPET smallRNA pipeline (https://biopet-docs.readthedocs.io/en/latest/pipelines/tinycap/) GENTRAP (Generic Transcriptome Analysis Pipeline, March 2017 https://git.lumc.nl/rig-framework/gentrap). Bowtie (version 1.1.1.) was used for reads alignment with the following settings (--seedmms 3 --seedlen 25 -k 3 --best --strata). RNAseq quality control (QC) outcomes are in Appendix A. miRNA read quantification is performed using HTseq-count version 0.6.1 andmiRNAs annotation was carried out with miRBase miRNA annotation (http://www.mirbase.org/) version 21miRDB. Normalization was carried out using the TMM (trimmed mean of M values). QC results are summarized in Appendix A. 

The DiffExpr2, an in-house developed pipeline, was used to identify differentially expressed miRNAs. Raw counts were normalized in EdgeR version 3.14 after fitting a generalized linear model. Differential expression was determined using the LRT (log-ratio likelihood). *p*-values were corrected for multiple testing using Benjamini and Hochberg’s corrections. miRNAs with a *p*-value < 0.05, false discovery rate (FDR) corrected, were considered significant (=differentially expressed). The lists of miRNAs in human and mouse RNAseq are found in Appendix A.

Gene targets for miRNAs were obtained from miRBase, only predicted targets, ≥80 target score, were included. From those, OPMD predicted targets were identified by overlapping with the OPMD downregulated gene list from [20].

### 4.3. qRT-PCR

RNA was converted to cDNA in the presence of spikes, sp3 and sp6, using the Exiqon’s miRCURY LNA Universal RT. Pipetting was carried out with a robot; PCR amplification was carried out with miRNAs specific primers and EXILENT Sybr green master mix in technical duplicates in a Roche light cycler 480 machine. CT values were made from the average of duplicates. CT > 38 was excluded. Expression levels were normalized to the average of sp3 and sp6, using the dCT calculation. In saliva, normalization was also to miR23a, since its product was amplified in all samples, and its levels were unchanged between control and OPMD. In serum, this step was omitted, since PCR product was not detected in all samples. When PCR products were obtained in only half of the control subjects or OPMD samples, it was excluded from the analysis.

### 4.4. Statistical Analysis

Correlation analyses were assessed with a Pearson test. A pFDR < 0.05 was considered as significant. Differences between qRT-PCR products were assessed with a Student’s *t*-test.

## Figures and Tables

**Figure 1 ijms-21-06059-f001:**
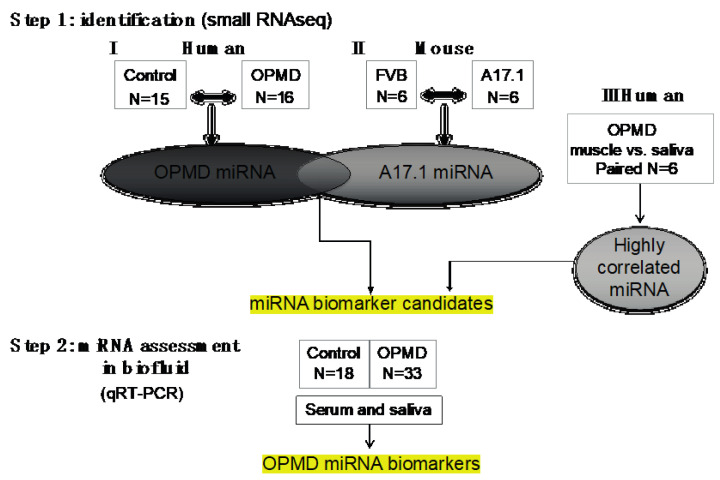
Workflow of our study. In step 1, the oculopharyngeal muscular dystrophy (OPMD)-miRNA candidates were identified in three RNAseq studies. (**I**) human control and OPMD, vastus lateralis (VL), (**II**) mouse OPMD model A17.1 and Friend Virus B laboratory mouse (FVB) control, tibialis anterior (TA). From both studies, differentially expressed (DE) miRNAs were identified. (**III**) A small RNAseq in muscle and saliva from OPMD patients was carried out in order to identify the highly correlated miRNAs between muscle and saliva. miRNA candidates were selected from the three studies. In step 2, assessment of expression was carried out using qRT-PCR in both serum and saliva from control and OPMD.

**Figure 2 ijms-21-06059-f002:**
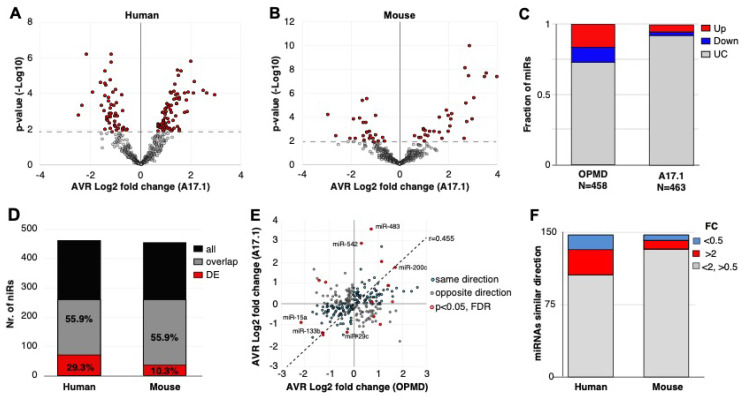
OPMD and A17.1 differentially expressed miRNAs. (**A**) and (**B**) Volcano plots show the average fold-change (log2) against *p*-value (-log10) in OPMD vastus lateralis (VL; **A**) or in A17.1 mouse tibialis anterior (TA; **B**). The miRNAs with pFDR < 0.05 are depicted in red. (**C**) Bar chart shows the fraction of deregulated miRNA in humans or mice. The upregulated miRNAs are depicted in red, and downregulated in blue. (**D**) Bar chart shows the number of total and overlapping miRNAs between mouse and human (depicted in black or grey, respectively). The proportion of overlapping miRNAs is from the total. The overlapping deregulated miRNAs are depicted in red, and the proportion of the deregulated miRNAs is from the overlapping miRNA pull. (**E**) Scatter plot of the average fold-change of the 252-overlapping miRNA. Pearson correlation coefficient (r) and regression line are depicted. Similar fold-change directions between human and mouse are depicted in cyan and the miRNA with opposite direction are in grey. The overlapping miRNAs with pFDR < 0.05 are encircled in red. The miRNAs with the highest or lowest fold-changes in both humans and mice are indicated. (**F**) Bar chart of the overlapping miRNAs with a similar fold-change direction in mouse and human. The number of miRNAs with a fold-change (FC) > 2 or < 0.5 are depicted in red and blue, respectively.

**Figure 3 ijms-21-06059-f003:**
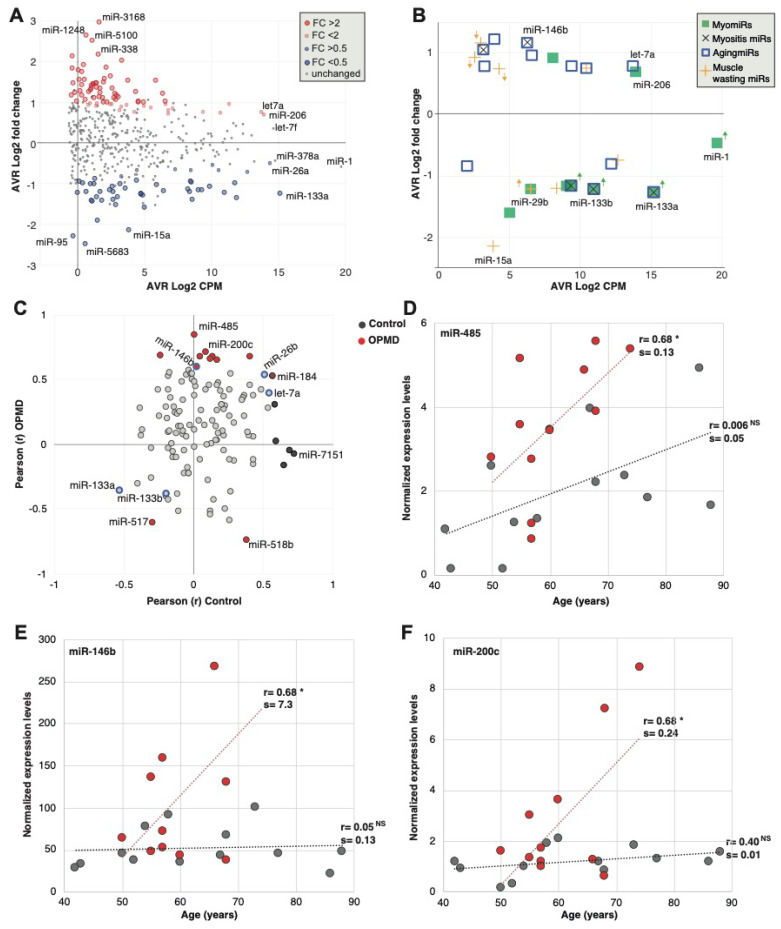
Analysis of miRNAs in human OPMD muscles. (**A**) Scatter plot shows average count per million (CPM, log 2) versus average fold-change (log2) in OPMD versus control muscles. The OPMD upregulated miRNAs are depicted in red, and the downregulated in blue, and those with fold-change > 2 or < 0.5 with a larger circle. The name of most abundant miRNAs and those with the highest fold-change is depicted. (**B**) Scatter plot shows average CPM versus average fold-change in OPMD for selected miRNAs that have been reported to be associated with several muscle pathologies and physiological conditions: The myomiRs are reported in muscular dystrophies; myositis miRs are reported in pathologies with muscle inflammation; AgingmiRs have been reported in aging muscles, and muscle wasting miRs have been reported in models with muscle atrophy and wasting. The miRNAs whose dysregulation direction differs between OPMD and the reference condition are marked with an arrow, and its color refers to the condition. Differences in dysregulation directions were found for muscular dystrophies and muscle wasting. (**C**) Scatter plot of the Pearson correlation coefficient (r) of miRNA expression levels with age in control and OPMD subjects. Analysis was carried out for the OPMD-deregulated miRNAs (pFDR < 0.05). The significantly correlated (*p* < 0.05) miRNAs in OPMD are depicted in red, and in control in black. The reported aging-associated miRNAs, miR-133a/b, miR-146b, miR-26b and let-7a are blue encircled. (**D**,**F**). Scatter plots of expression values versus age in three significantly age-correlated and OPMD-deregulated miRNAs. Controls are depicted in grey and OPMD in red. A dotted line shows a linear regression line, the correlation coefficient (r) is denoted. Pearson *p*-value < 0.05 is denoted *, respectively, or non-specific (NS). The slope (=s) of the regression line was determined with a linear regression model.

**Figure 4 ijms-21-06059-f004:**
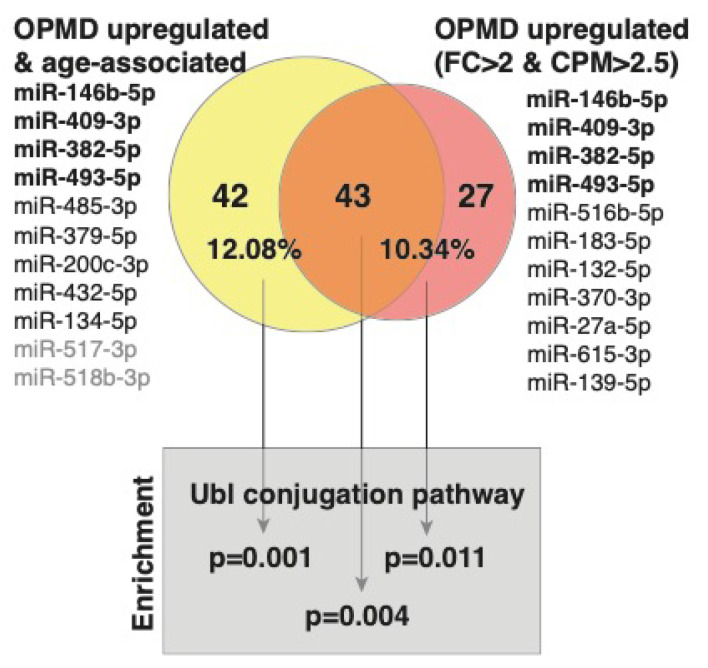
A graphical summary of predicted targets analysis. The OPMD upregulated miRNAs and those that are also age-associated are listed. The OPMD upregulated miRNAs inclusion criteria: average fold-change (FC) > 2 and average CPM > 2.5. In bold are the four miRNAs that are common between the two conditions. The miRNAs without predicted gene targets are in grey. Venn diagram shows the overlap between the number of OPMD-downregulated predicted targets to the OPMD upregulated miRNAs that are age-associated (yellow) and the OPMD upregulated miRNAs (**red**). The percentage of predicted targets from the OPMD downregulated gene list is depicted.

**Figure 5 ijms-21-06059-f005:**
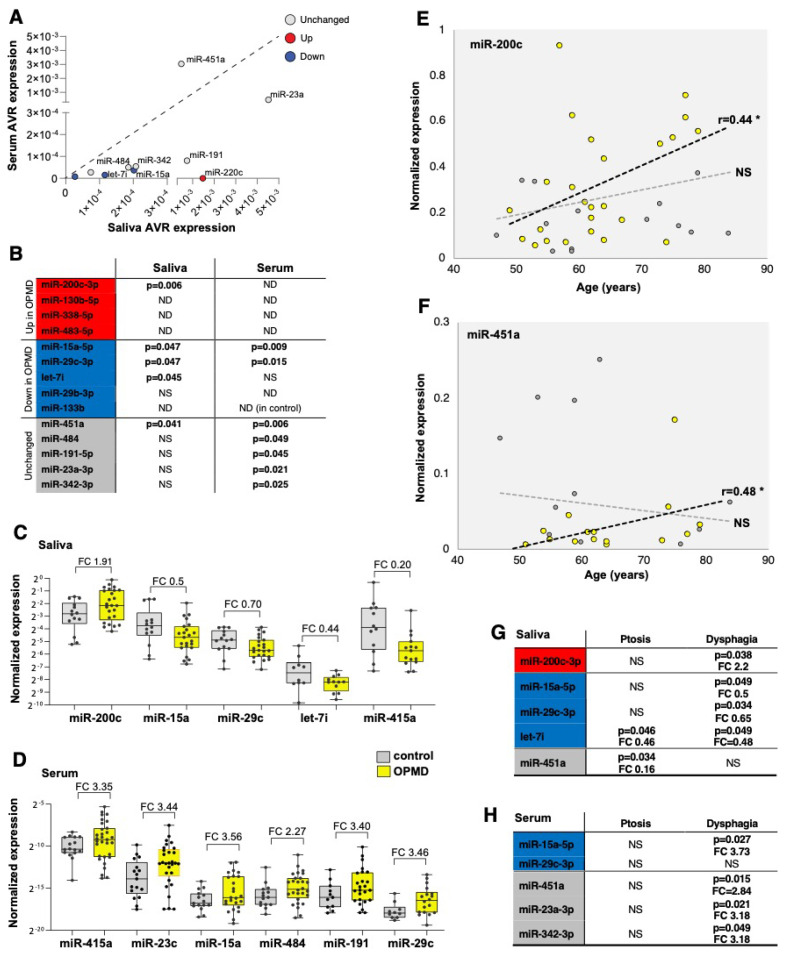
miRNA candidates for OPMD in serum and saliva. (**A**) Scatter plot shows average expression levels in saliva vs. serum. The dysregulation direction (up or down in OPMD muscles) is denoted in red or blue, respectively. miRNAs with unchanged levels in muscles are denoted in grey. The diagonal (1:1 ratio) is denoted with a dashed line. (**B**) A summary qRT-PCR for 14 miRNAs in serum or saliva. *p*-value (*p* < 0.05) indicates a significant difference between control and OPMD. miRNAs that are higher in OPMD muscles are in red, those with lower expression levels in blue, and unchanged in grey. *p*-values were determined with an unpaired *t*-test, Welch corrected. ND, undetected; NS, not significant. miR-133b was not detected in serum from controls. (**C**,**D**) Box plots show expression levels in saliva (**C**) and serum (**D**) in control and OPMD groups of the significantly differentially expressed miRNAs. Fold-change (FC) is depicted. Every subject is denoted with a dot. The fold-change of all the differentially expressed miRNAs in serum has an opposite direction than in OPMD muscles. (**E**,**F**) An age-associated expression level is visualized with a scatter plot. Significant correlation (Pearson, *p* < 0.05, denoted with *) was found only for in miR-200c and miR-451a in saliva, and only for the OPMD samples. NS = not significant. Correlation coefficient (r) is denoted next to the regression line (dashed lines). A thin dashed line denoted the regression line of the controls and the bold line denotes the OPMD samples. (**G**,**H**) A summary of miRNA differential expression analysis per initial symptom, ptosis (*n* > 12) or dysphagia (*n* > 16). The *p*-value (unpaired *t*-test) and fold-change (FC) are denoted for the significantly differentially expressed miRNAs. miRNAs that are higher in OPMD muscles are in red, those with lower expression levels in blue, and unchanged in grey.

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
