# Peer review of "Age-Associated Salivary MicroRNA Biomarkers for Oculopharyngeal Muscular Dystrophy"

_ijms, 2020, doi:10.3390/ijms21176059_

Round 1

Reviewer 1 Report

Raz et al. report microRNA biomarkers from muscles, blood and saliva of human OPMD patients and mouse model. They identified 15 OPMD-deregulated miRNAs overlapped between both and the majority of these deregulated miRNAs showed opposite direction to reported muscular dystrophy-associated miRNAs, however the similar direction of miRNAs was found between OPMD and aging muscles. Furthermore, they also analyzed the deregulation of miRNAs in biofluids. There are some major concerns in this submitted version of the manuscript.

  1. In this paper, the authors showed good correlation of the expression of miRNAs with the age or with initial symptoms. However, the purpose for identification of OPMD-related miRNAs as biomarkers is not clear. As mentioned in introduction, if the authors want to establish biomarkers for development of therapies, they should show the correlation between the miRNA expression and clinical information (symptoms) on the progression of disease, but not only with age. If their aim is to establish biomarkers for diagnosis, they should show the specificity of the expression in comparison with those in the other muscle diseases more comprehensively.
  2. In Figure 3D-F and 5 C and D, there is a large variation among the patients on the expression of miRNAs in all of materials, muscles, saliva and blood. Even they showed some significance in average, the data of individual patient will be very difficult to be evaluated on practical use as biomarkers. How do we use the data from individual patients? Please discuss it.
  3. In discussion, the authors claimed the values in saliva might be more reliable by showing the same direction to those in muscles. However, there are no data for the reproducibility in measurement of saliva as the concentration will be varied in sampling time, as I concern that the values will be influenced by saliva production. Can you show the data?
  4. Please discuss why the data in blood are opposite to those in muscles and saliva. This is strange.
  5. The fourth paragraph in discussion is almost nothing. They discussed the expression of miR200c in AD as a protein aggregation disease. But the pathogenesis and protein aggregation mode in both of AD and OPMD are so different. And suddenly they changed the discussion in that the miR200c will be related to muscle wasting and myogenesis. As muscle wasting and muscle regeneration are hallmarks in almost all muscle diseases, including muscular dystrophies. Please focus on the point in Discussion and make it with the similarities and differences of the expression to those in other muscle diseases.
  6. The labels in Figures are sometimes mistaken. For example, in Figure 4.  All of common miRNAs are not in bold (miR382-5p, 493-5p). In Fig5A and C, E, miR220c should be miR200c, mIR415a should be 451a. Carefully check them.

Author Response

Reviewer 1:

We thank the reviewer for the constructive comments that helped us to revise the manuscript.

  1. In this paper, the authors showed good correlation of the expression of miRNAs with the age or with initial symptoms. However, the purpose for identification of OPMD-related miRNAs as biomarkers is not clear. As mentioned in introduction, if the authors want to establish biomarkers for development of therapies, they should show the correlation between the miRNA expression and clinical information (symptoms) on the progression of disease, but not only with age. If their aim is to establish biomarkers for diagnosis, they should show the specificity of the expression in comparison with those in the other muscle diseases more comprehensively.

We thank the reviewer for this comment requiring a clarification of study objectives. This is now better described in the Abstract and Introduction of the revised version. In brief, the clinical diagnostics for OPMD is based on neurological tests that are carried out by a neurologist. Quantitative biomarkers can report an objective diagnosis. A correlation between miRNA levels and initial symptoms are presented in Figure 5G-H. To measure progression, we should establish a longitudinal study, which is not part of the ethical document in this study. Here we assessed only investigated an age-association. We agree with the reviewer that assessing disease progression is more accurate. Progression is depicted from a longitudinal study, whereas here we used cross-sectional studied. A longitudinal study requires a new ethical approval. To avoid confusion, we rewrote all sections and pointing that we assessed an age-association (instead of progression). The textual changes are found in lines: 37, 58-60, 64, 76-83, 444, 457-461.

  1. In Figure 3D-F and 5 C and D, there is a large variation among the patients on the expression of miRNAs in all of materials, muscles, saliva and blood. Even they showed some significance in average, the data of individual patient will be very difficult to be evaluated on practical use as biomarkers. How do we use the data from individual patients? Please discuss it.

We agree with the reviewer that there is a high inter-patient variation. This could be expected considering the high variations in symptom severity between OPMD patients. We think that few miRNAs would be required for proper diagnostics of OPMD. Additional studies should be performed to verify our results but focusing on a short list of only several miRNAs. From those studies together, a range of expression levels should made for control (unaffected/not progressed) and OPMD. This is how it is done in other pathologies. Since this is the first miRNA study for OPMD, we feel that it is too early for specific recommendation. In agreement with the reviewer we added text in the discussion, lines: 489-563.

  1. In discussion, the authors claimed the values in saliva might be more reliable by showing the same direction to those in muscles. However, there are no data for the reproducibility in measurement of saliva as the concentration will be varied in sampling time, as I concern that the values will be influenced by saliva production. Can you show the data?

miRNAs, in general, are highly stable and therefore are used as biomarkers.

Our conclusion that miRNA levels in muscles is better captured in saliva compared with serum is based on two studies: 1- in the RNAseq study shows a correlation in expression levels between muscles and saliva for only a subset of miRNAs (suppl Fig. 2). 2- for all the selected miRNAs, but one, in average higher expression levels were found in saliva vs. serum across all subjects (controls and OPMD). If expression levels are affected by saliva production, and this differs per subject, we would not find higher expression levels in saliva compared with serum. We agree that this is one measurement, but it holds also for the serum. We agree with the reviewer that a longitudinal study should be followed. This is indicated in lines: 460-461.

  1. Please discuss why the data in blood are opposite to those in muscles and saliva. This is strange.

miRNAs are secreted and the expression levels in both biofluids could be affected by other tissues. A different study should design to address this question, it is beyond the scope of this study. Without an experimentally supported scientific answer, we would not like to speculate as to the differences in expression direction in this paper.

  1. The fourth paragraph in discussion is almost nothing. They discussed the expression of miR200c in AD as a protein aggregation disease. But the pathogenesis and protein aggregation mode in both of AD and OPMD are so different. And suddenly they changed the discussion in that the miR200c will be related to muscle wasting and myogenesis. As muscle wasting and muscle regeneration are hallmarks in almost all muscle diseases, including muscular dystrophies. Please focus on the point in Discussion and make it with the similarities and differences of the expression to those in other muscle diseases.

In agreement with the reviewer’s comment this paragraph has been rewritten focusing on miR-200c effect on muscle wasting, myogenesis and protein aggregation. All cellular and tissue defects are attributed to OPMD.  The revised text is found in lines: 473-477.

  1. The labels in Figures are sometimes mistaken. For example, in Figure 4.  All of common miRNAs are not in bold (miR382-5p, 493-5p). In Fig5A and C, E, miR220c should be miR200c, mIR415a should be 451a. Carefully check them.

We apologize for the typing errors in the figures, those have been fixed, and the figure ledged for Fig. 4 has been rewritten.

Reviewer 2 Report

Research has clinical and scientific value. There are several weaknesses in the article that authors need to complete. Please provide a detailed description of the collection of saliva, criteria for inclusion and exclusion of patients from the examination as well as the description and results of the dental examination. Since saliva was the research material, a description of the clinical condition of the oral cavity must be provided. Of course, the composition of saliva is influenced by systemic diseases, but most of all by the clinical condition of the oral cavity !!!!!

Author Response

Reviewer 2:

We thank the reviewer for the constructive comments that helped us to revise the manuscript.

  1. Methods and results can be

We have rewritten parts the methods and Results sections based on the specific comments of reviewer 1.

  1. Please provide a detailed description of the collection of saliva, criteria for inclusion and exclusion of patients from the examination as well as the description and results of the dental examination. Since saliva was the research material, a description of the clinical condition of the oral cavity must be provided. Of course, the composition of saliva is influenced by systemic diseases, but most of all by the clinical condition of the oral cavity !!!!!

We wrote parts of the Methods in more details and included the inclusion criteria (lines 565-580).

Inclusion criteria:

    • Only subjects older than 18 years were included.
    •  OPMD patients were clinically diagnosed by the neurologist based on the common clinical OPMD symptoms. OPMD was genetically confirmed using a genetic test as described in Raz et al., 2019. Patients with OPMD-like symptoms that were not genetically confirmed were excluded.
    • Healthy controls were the partners of the OPMD patients.
  • No inclusion or exclusion criteria were based on the oral cavity or dental examination. Dental or oral cavity problems have not been reported in OPMD patients. We cannot add this information into the study as it is not included in the ethical form for this study.

Round 2

Reviewer 2 Report

I do not agree that patients with OPMD have no problems with the gums or caries. Please mark this as a weak point of the publication